

# Derivative-free optimization adversarial attacks for graph convolutional networks

Runze Yang and Teng Long

School of Information Engineering, China University of Geosciences, Beijing, China

## ABSTRACT

In recent years, graph convolutional networks (GCNs) have emerged rapidly due to their excellent performance in graph data processing. However, recent researches show that GCNs are vulnerable to adversarial attacks. An attacker can maliciously modify edges or nodes of the graph to mislead the model's classification of the target nodes, or even cause a degradation of the model's overall classification performance. In this paper, we first propose a black-box adversarial attack framework based on derivative-free optimization (DFO) to generate graph adversarial examples without using gradient and apply advanced DFO algorithms conveniently. Second, we implement a direct attack algorithm (DFDA) using the Nevergrad library based on the framework. Additionally, we overcome the problem of large search space by redesigning the perturbation vector using constraint size. Finally, we conducted a series of experiments on different datasets and parameters. The results show that DFDA outperforms Nettack in most cases, and it can achieve an average attack success rate of more than 95% on the Cora dataset when perturbing at most eight edges. This demonstrates that our framework can fully exploit the potential of DFO methods in node classification adversarial attacks.

# INTRODUCTION

Graph convolutional networks (GCNs) (*Kipf & Welling, 2017*) are one of the most popular graph neural networks (GNNs) (*Scarselli et al., 2009*). They are widely used in recommendation systems (*He et al., 2020*), molecular chemistry (*Ryu, Kwon & Kim, 2019*) and knowledge graphs (*Wang et al., 2019*). Like traditional neural networks, GCNs are also vulnerable to adversarial attacks. The adversary can modify structure and node features to make the model misclassify the target nodes (*Zügner, Akbarnejad & Günnemann, 2018*; *Dai et al., 2018*; *Ma et al., 2019*), or even cause the overall classification performance to degrade (*Zügner & Günnemann, 2019*). Therefore, the study of GCNs' robustness has received increasingly widespread attention (*Sun et al., 2018*).

Gradient-based adversarial attacks in the continuous domain have had a series of related works on both images (*Goodfellow, Shlens & Szegedy, 2015*; *Papernot et al., 2016*) and audio (*Carlini & Wagner, 2018*). Unlike adversarial attacks in the continuous domain, graph data are often discrete: the structural information (adjacency matrix) of the graph is discontinuous and the features may also have discrete values. This makes it difficult to use gradient information to attack (*Zügner, Akbarnejad & Günnemann, 2018*),

Corresponding author
Teng Long, longteng@cugb.edu.cn

especially under a black-box condition where only classification output vector can be obtained.

Derivative-free optimization (DFO) algorithms (*Conn, Scheinberg & Vicente, 2009*) are a class of algorithms that do not compute the gradient but only use the value of the objective function to optimize. These algorithms are often used in cases where the derivative of the objective function is undefined, or where it is difficult to obtain a reliable value of the derivative. It has been successfully applied to attack traditional deep neural networks (*Ughi, Abrol & Tanner, 2020*). There is also some DFO attack work on GNNs. For example, *Dai et al. (2018)* implemented a black-box GCN adversarial attack algorithm based on a genetic algorithm by setting the population, fitness, selection, crossover and mutation in detail. *Chen et al. (2019)* proposed a community detection attack algorithm with a genetic algorithm and verified their algorithm has good transferability. However, without a uniform framework, these works usually have to implement custom versions of the algorithms for a certain problem. To the authors' best knowledge, there is no general framework that can quickly apply various DFO algorithms in the field of graph adversarial attacks.

In this paper, we propose a black-box adversarial attack framework based on the idea of DFO. It consists of three steps: *Input Setting* (design the loss function, perturbation vector, constraints and so on), *Iterative Query* (generate perturbation vectors and query the black-box GCN model iteratively) and *Final Perturbation* (modify graph data with perturbation that minimize the loss function). In facing the difficulty of using gradient and the inconvenience of applying and comparing DFO algorithms, the key idea insight of our approach is (1) regarding graph adversarial attacks as a search problem in a discrete solution space and using derivative-free optimizers (DFOers) to solve it; (2) abstracting the specific task of graph adversarial attacks as an optimization problem about a certain form perturbation vector in order to switch and compare various DFOers conveniently.

Moreover, we use the Nevergrad (*Rapin & Teytaud, 2018*) library to implement a black-box direct adversarial attack algorithm (called DFDA) on GCN-based node classification tasks. Following the framework above, we set attack loss function, perturbation vector, perturbation constraint, mapping function and derivative-free optimizer separately.

We conducted a series of experiments on Cora, Citeseer and Polblogs. Without loss of generality, we attack node 0 of the Cora dataset with five different DFOers to compare the classification margin and comprehensive performance. Then, we randomly select 50 nodes to attack separately to study the average attack success rate of DFDA with different iteration numbers, perturbation constraints and perturbation types. Finally, we compare DFDA with a classical algorithm Nettack (*Zügner, Akbarnejad & Günnemann, 2018*)—a well-performing greedy algorithm—under different defense models. *Nettack* is a well-performing adversarial attack algorithm based on greedy approach. The results show that all the selected DFOers can search for effective perturbations, and DFDA is superior to Nettack in most cases.

The contributions of this paper are listed below:

- We have proposed a black-box adversarial attack framework in order to generate graph adversarial samples without using the gradient. Specifically, we use DFO methods to perform attacks effectively. Additionally, the uniform outputs of DFOers (perturbation vectors) make it convenient to use and compare different DFO algorithms.

- We have implemented a direct adversarial attack algorithm on node classification tasks for GCNs based on the framework above. Facing the potential problem of too large search space, we set the perturbation vector dimension to the constraint size and set the elements of the perturbation vector as the pointers indicating the perturbed position of the original matrix. This approach reduces the search space from exponential level to power level and enables the perturbation vector to pass the constraint check more easily.

- We have conducted a series of experiments under various conditions. The results show that we can achieve an average attack success rate of more than 95% on the Cora dataset when perturbing at most eight edges. We compare our algorithm with the classical algorithm *Nettack* under different defense models and find that DFDA outperforms *Nettack* in most cases. During the experiments, instead of copying an original graph at each perturbation iteration, we use inverse perturbation to restore the perturbed graph to its original state. This can effectively reduce the computation cost under a large number of iterations.

This paper is organized as follows "Preliminaries" gives the basic concepts of GCN-based node classification and adversarial attacks. "Derivative-Free Adversarial Attack on GCNs" describes our framework and algorithms, followed by the experimental results in "Evaluation". "Related Work" introduced the related work, followed by some concluding remarks in "Conclusion".

## RELATED WORK

There has been some classical work on node classification adversarial attacks. The concept of graph adversarial attacks was first introduced by *Zügner, Akbarnejad & Günnemann (2018)*. They proposed a gray-box attack algorithm Nettack based on the greedy approach. In this algorithm, the attacker can obtain training labels to train the surrogate model. The adversary can generate an adversarial sample by attacking the surrogate model and migrate it to the target model for attacks. Subsequently, they proposed *Mettack* (*Zügner & Günnemann, 2019*), an attack algorithm that can reduce the global classification accuracy. This algorithm attacks based on the gradient information of the adjacency matrix. *Dai et al. (2018)* proposed a black-box attack algorithm named *RL-S2V* based on reinforcement learning. This method performs attacks by injecting nodes into the graph.

Nettack is a global attack algorithm that attacks to decline the classification accuracy of a whole graph instead of a target node. Black-box queries in *RL-S2V* setting only return the prediction classes rather than class probability vectors. Therefore we do not compare

with the above algorithms. We compare DFDA with Nettack because the attack settings are more similar: both of them can perform direct perturbations of edges and features of a target node. For the sake of reasonableness, we unify Nettack's settings in terms of node selection and constraints with DFDA.

DFO methods like genetic algorithms are used in some graph adversarial attack tasks. For example, *Dai et al. (2018)* applied genetic algorithms to node classification adversarial attacks and adapted it as a baseline algorithm for their main algorithm RL-S2V. Experiments in their paper demonstrate that genetic algorithms are effective for combinatorial optimization problems like graph adversarial attacks. *Chen et al. (2019)* proposed a community detection attack algorithm based on genetic algorithms and verified that the algorithm has good transferability. In this paper, we propose an adversarial attack framework to make it convenient to apply and compare different DFO algorithms.

Nevergrad (*Rapin & Teytaud, 2018*) is a Python3 based open-source framework developed by Facebook, which provides a large number of implementations of DFO algorithm optimizers, such as, differential evolution algorithms, fast genetic algorithms, covariance matrix adaptive algorithms, particle swarm optimization algorithms, etc. When DFO algorithms need to be applied, researchers usually need to implement custom versions of algorithms for real problems, which consume a lot of time and do not facilitate the comparisons among algorithms. The algorithm DFDA we proposed is based on Nevergrad, using which the strengths and weaknesses of various DFO algorithms can be easily compared. It can help researchers find suitable algorithms that deserve to be further customized.

## PRELIMINARIES

In this section, we introduce the notions about GCN-based node classification and the general form of adversarial attack on graph.

### Node classification with GCN

The classification task on graph data mainly consists of two cases, one is graph-level classification, where the graph $G$ is a whole with a label, and the other is node-level classification, where each node in the graph $G$ belongs to a class in the label set $Y$. In this paper, we focus on node-level semi-supervised classification. Semi-supervised node classification is a task in which the labels of unknown nodes are derived by training under the condition that the training set labels are known.

In the following we give the mathematical definition of semi-supervised node classification. For an undirected graph $G$, given the adjacency matrix $A \in \{0,1\}^{N \times N}$, the node feature matrix $X \in \mathbb{R}^{N \times d}$ (the feature of the node $i$ is $x_i \in \mathbb{R}^d$), the label of the $i$-th node $y_i \in \{1, 2, \ldots, k\}$, a set of labeled nodes $V_L$ (training set, where $|V_L| = N_L$), and a set of unlabeled nodes $V_U$ (test set, where $|V_U| = N_U$). The objective is to train a model $f_\theta(G)$ on the graph $G$ with training parameter $\theta$ that predicts the label of each node in $V_U$.

The general idea of semi-supervised learning model training is to minimize the loss function of the model on the training set as much as possible, *i.e.*,

$$\min \mathscr{L}_{train}(f_\theta(G)) = \sum_{v_i \in V_L} \ell\big(f_\theta(A, X)_i, y_i\big) \tag{1}$$

where $f_\theta(A, X)_i$ denotes the class probilities of node $v_i$ and $y_i$ denotes the true label of node $v_i$. The loss function $\ell(\cdot, \cdot)$ denotes the cross-entropy error.

We use the 2-layer GCN proposed by *Kipf & Welling (2017)* as the target model. The 2-layer GCN is one of the victim models commonly used in adversarial attack experiments. It is defined as follows:

$$f(\mathbf{A}, \mathbf{X}) = Softmax\Big(\hat{\mathbf{A}} ReLU\big(\hat{\mathbf{A}} \mathbf{X} \mathbf{W}^{(1)}\big) \mathbf{W}^{(2)}\Big) \tag{2}$$

where $\hat{\mathbf{A}} = \widetilde{\mathbf{D}}^{-\frac{1}{2}} \widetilde{\mathbf{A}} \widetilde{\mathbf{D}}^{-\frac{1}{2}}$, $\widetilde{\mathbf{A}} = \mathbf{A} + \mathbf{I}$ and $\widetilde{\mathbf{D}}_{ii} = \sum_j \widetilde{\mathbf{A}}_{ij}$. Each row vector $f(\mathbf{A}, \mathbf{X})_i \in [0,1]^k$ of the output matrix represents the class probability vector of node $v_i$.

### General forms of adversarial attack

Given a graph $G = (A, X)$ and a set of attack target nodes $V_t \subseteq V$. Let $y_i$ denote the true class of node $v_i$. For node $v_i$ in the test set, since we do not know its true label, we can adopt the method of self-learning (*Zügner & Günnemann, 2019*), that is, regarding the model output on the clean graph as the true labels of unknown nodes. The attack objective is to find a perturbed graph $G' = (A', X')$, which makes the attack target function $\mathscr{L}_{atk}$ minimum, *i.e.*,

$$\min \mathscr{L}_{atk}\big(f_\theta(G')\big) = \sum_{v_i \in V_t} \ell_{atk}\Big(f_{\theta^*}(G')_i, y_i\Big)$$

$$\text{s.t.,} \quad \theta^* = \arg\min_\theta \mathscr{L}_{train}\big(f_\theta(\hat{G})\big) \tag{3}$$

where $\ell_{atk}$ is the loss function of the attack. $\hat{G}$ can be chosen as either the original graph $G$ or the perturbed graph $G'$, corresponding to the poisoning attack scenario and the escape attack scenario, respectively. Poisoning attack means that the GCN will be retrained with the perturbed graph while evasion attack represents the cases that the GCN will not be retrained (*Sun et al., 2018*).

In particular, for an adversarial attack with a certain single node $v_i$, the objective function can be formulated as:

$$\min \mathscr{L}_{atk}\big(f_\theta(G')\big) = \ell_{atk}\Big(f_{\theta^*}(G')_i, y_i\Big). \tag{4}$$

However, the perturbation of $G'$ is constrained and does not allow unrestricted modification on the graph. A realistic assumption is that the attack needs to generate as small and indistinguishable perturbations as possible, *i.e.*, $G \in \phi(G)$, where $\phi(G)$ denotes the constraint domain. If a perturbation upper bound $\Delta$ is given, a typical perturbation constraint can be expressed as

$$\|A - A'\|_0 + \|X - X'\|_0 \leq \Delta. \tag{5}$$

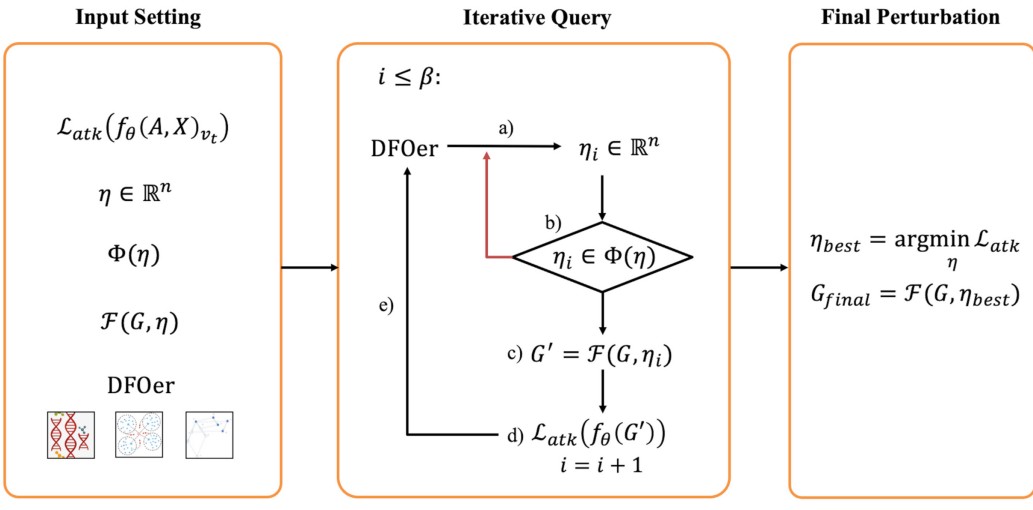

**Figure 1 Derivative-free black-box attack framework.**

# DERIVATIVE-FREE ADVERSARIAL ATTACK ON GCNS

In this section, we introduce a black-box adversarial attack framework for GCNs based on the idea of DFO. And then, we implement a direct attack algorithm on the GCN node classification task using the *Nevergrad* algorithm library. Finally, we optimize the attack algorithm to solve the problem of excessive dimensionality encountered during the algorithm implementation.

## Derivative-free black-box attack framework

The core idea of the framework is mainly twofold: (1) we treat graph adversarial attacks as a search problem in a discrete solution space and use derivative-free optimizers (DFOers) to solve it; (2) we abstract the specific task of graph adversarial attacks as an optimization model about perturbation vectors. The former mainly copes with the problem of difficult gradient utilization in black-box graph adversarial attacks. The DFOers can optimize without computing the gradient. The latter reflects the generality of the framework. The framework unifies the output of the DFOer into a certain form perturbation vector, which provides a basis for the rapid switching and comparison of various DFOers.

The main elements of the framework are described below. This framework contains three main steps: *Input Setting*, *Iterative Query* and *Final Perturbation*. The schematic diagram of the framework is shown in Fig. 1.

### Input setting

In this step, we need to define the attack loss function $\mathscr{L}_{atk}$, perturbation vector $\eta$, perturbation constraint $\phi(\eta)$, mapping function $\mathscr{F}$ and the DFOer *Opt*. In the black-box scenario, we only know the adjacency matrix, the feature matrix and the black-box query result in the form of class probability vectors. So the attack loss function is usually constructed from the black-box query results. It is the target of our optimization. In general, the smaller the function value, the better the attack effect. The perturbation vector

is the interface between the DFOer and the loss function. The uniform perturbation vector ensures the feasibility of switching the DFOer for comparison experiments. The perturbation constraint defines the restrictions on the perturbation vector generated by the DFOer. The mapping function defines how the perturbation is imposed on the graph $G$. The DFOer can return a perturbation vector that tends to make the loss function as small as possible by learning the value of the loss function obtained from the query.

### Iterative query

After defining the input, we will perform $\beta \in \mathbb{N}$ iterations to generate the candidate perturbation vectors. Within the number of iterations $\beta$ (also called resource budget), repeat: (a) generate a perturbation vector with the DFOer; (b) check whether the perturbation vector violates the constraints and regenerate the vector that violates them; (c) impose the perturbation to the original graph $G$ to generate the perturbed graph $G'$; (d) input the perturbation graph $G'$ into the GCN model and query the attack loss function value; (e) return it to the optimizer. In this step, all perturbation vectors and the corresponding loss values are recorded.

### Final perturbation

At the end of the iterative query, the perturbation vector that minimizes the loss function will be imposed to the graph $G$ to generate $G_{final}$. At this point, the whole adversarial sample generation process is finished. Subsequently, we put $G_{final}$ in (3) as $G'$ for poisoning attacks or evasion attacks.

## Derivative-free direct attack (DFDA)

The Derivative-Free Direct Attack (DFDA) algorithm is a black-box adversarial attack method on GCN node classification tasks. In this algorithm, the adversary can directly modify the connections and features of the target node to mislead GCNs to misclassify the node as a chosen class (called target class). DFDA can perform structural perturbation (modifying the adjacency matrix) and feature perturbation (modifying the feature matrix).

First, we set up the five subsections of *Input Setting* of the framework.

### Attack loss function

Considering the success rate of the attack, the second most probable class originally predicted by the clean graph is selected as the target class. The loss function is designed as,

$$\mathcal{L}_{atk} = f(\boldsymbol{A}, \boldsymbol{X})_{v_t, c_1} - f(\boldsymbol{A}, \boldsymbol{X})_{v_t, c_2} \tag{6}$$

$f(\boldsymbol{A}, \boldsymbol{X})_{vt}$ is the output of the black-box query of the target node $v_t$ as a class probability vector. $c_1$ and $c_2$ denote the first and second largest probability classes of the black-box query before the attack. When the loss function decreases, the target class probability increases and the correct class probability decreases.

### Perturbation vector

For structural perturbations, the perturbation vector needs to be set to $\{0, 1\}^{N-1}$ to describe all possible perturbations that can be generated, since the target node may have

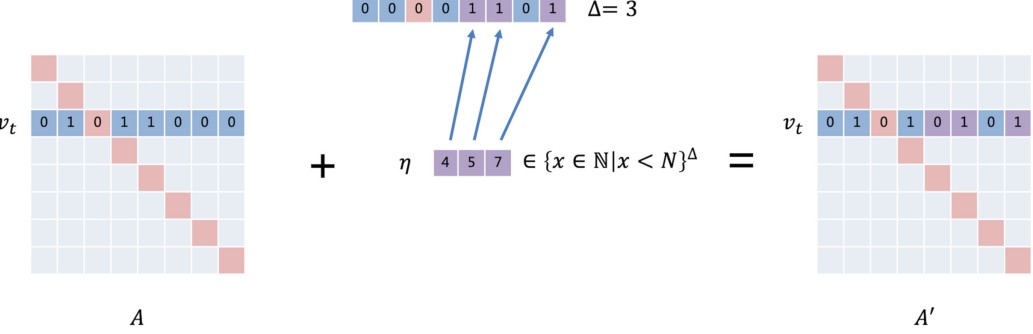

**Figure 2 The setting of the perturbation vector.** The upper vector is in the form of $\{0,1\}^N$. 0 means that the connection status remains unchanged, and 1 means that theedge is added or removed. We designed the lower vector, which uses the position pointers as elements to decline the dimension.

edges with all other $N-1$ nodes. The size of the search space for this problem is $O(2^N)$, which is of exponential level. Since the number of edges that an attacker can modify (usually set to $\Delta$) is very small to ensure the invisibility of the perturbation in practical situations, the perturbation constraint is usually very tight. The vast majority of perturbations are not qualified. To reduce the search space and improve the probability of passing the constraint check, we define the structural perturbation vector as

$\eta_A \in \{x \in \mathbb{N} | x < N\}^\Delta$. As is shown in Fig. 2, each element $u \in \eta_A$ like a position pointer represents that the connection between the target node and the node $v_u$ is changed. This method reduces the search space significantly (down to $O(N^\Delta)$). Similarly, we set the feature perturbation vector to $\eta_X \in \{x \in \mathbb{N} | x < d\}^\Delta$. Here, we only consider the case where the feature matrix is of the form $X = \{0,1\}^{N \times d}$.

### Perturbation constraint

Based on the design of the above perturbation vector, we define the constraints as: (a) no duplicate elements in $\eta_A$ and $\eta_X$; and (b) $v_t \notin \eta_A$. Constraint $a$ guarantees that no duplicate modifications will be made to a particular edge while $b$ guarantees that the graph structure will not generate self-loops.

### Mapping function

The perturbation mapping function is defined as follows.

For each element $i$ in $\eta_A$,

$$A_{v_t i} := 1 - A_{v_t i}. \tag{7}$$

For each element $j$ in $\eta_F$,

$$X_{v_t j} := 1 - X_{v_t j}. \tag{8}$$

### Derivative-free optimizer

We choose OnePlusOne, DiscreteOnePlusOne, DoubleFastGAOnePlusOne, DE and RandomSearch in *Nevergrad* as DFOers to generate the perturbation vectors.

After defining the output, we follow the framework for *Iterative Query* and *Final Perturbation*. In the *Iterative Query* step, our approach is slightly different from the framework. In each iteration, we let the DFOer generate three perturbation vectors simultaneously, perform constraint checking and loss query separately, and finally select the perturbation vector with the smallest attack loss as the output of this iteration. These can make the optimization process more stable.

# EVALUATION

We have implemented our algorithm on DeepRobust (*Li et al., 2020*), an adversarial attack algorithm library developed on PyTorch. As DeepRobust integrates classical attack and defense algorithms on the image and graph domains, it can support the comparison between our algorithm and other algorithms.

We consider the following three research questions:

- **RQ1:** Which DFOer is the most suitable in the setting of DFDA?
- **RQ2:** To what extent does parameters like constraint size affect the attack results?
- **RQ3:** Compare with *Nettack*, how does our method perform under different defense models and scenarios?

To answer these questions, we design three kinds of experiments in the next section.

To answer RQ1, we conducted a single-node experiment, attacking only one node, to analyze the attack results. Without loss of generality, we compare the effect of different DFOers on node 0 in Cora. In this experiment, we select the most appropriate optimizer for subsequent experiments.

To answer RQ2, we conducted multi-node experiments in which all nodes in a target node set were attacked separately. The percentage of "successful" and "misleading" nodes were counted. In the multi-node experiments, we investigate the effects of resource budget $\beta$, perturbation constraint $\Delta$, and perturbation type on the success rate of the attack.

To answer RQ3, we conducted comprehensive attack experiments on different datasets and different defense models. In these experiments, we investigate the attack effect of DFDA under three defense models in evasion attack and poisoning attack scenarios. We also compare the DFDA with the greedy algorithm Nettack in poisoning attack cases.

# DATASET AND SETTINGS

## Dataset

The commonly used datasets in the field of graph adversarial attacks are Cora, Citeseer and Polblogs (*Rossi & Ahmed, 2015*). We present the statistics for each dataset in the following Table 1. Among them, Cora and Citeseer are attribute graphs, *i.e.*, each node in the graph has a specific dimensional attribute; the Polblogs dataset is a directed weighted graph with no node features. Cora and Citeseer are more sparse, while Polblogs is denser and has only two categories. Due to their strong representation, we perform attacks based on these three datasets.

---

**Algorithm 1** The derivative-free direct attack (DFDA) algorithm.

---

**Input:** Original clean graph $G = (A, X)$, target node $v_t \in V t$, resource budget $\beta$, perturbation constraint $\delta$, black-box target model $\mathcal{M}$

**Output:** Perturbed adversarial sample $G = (A', X')$

1: $p_{v_t} \leftarrow Query(\mathcal{M}, v_t)$

2: Get the most probable class $c_1$ and the second most probable class $c_2$ in $pv_t$

3: $t \leftarrow 0$

4: **while** $t < \beta$ **do**

5: **repeat**

6: Generate perturbation vector $\eta_A^{(t)} \in \{x \in \mathbb{N} | x < N\}^\Delta, \eta_X^{(t)} \in \{x \in \mathbb{N} | x < d\}^\Delta$ with a chosen DFOer

7: **until** $v_t \notin \eta_A^{(t)}$ $\eta_A^{(t)}, \eta_X^{(t)}$ does not contain duplicate elements

8: $\mathbf{A}^{(t)} \leftarrow \mathbf{A}, \mathbf{X}^{(t)} \leftarrow \mathbf{X}$

9: **for** each element $i$ in vector $\eta_A^{(t)}$ **do**

10: $\mathbf{A}^{(t)}[v_t][i] \leftarrow 1 - \mathbf{A}[v_t][i]$

11: **end for**

12: **for** each element $j$ in vector $\eta_X^{(t)}$ **do**

13: $\mathbf{X}^{(t)}[v_t][i] \leftarrow 1 - \mathbf{X}[v_t][i]$

14: **end for**

15: $p_{v_t} \leftarrow Query(\mathcal{M}, v_t)$

16: $loss^{(t)} \leftarrow p_{v_t}^{(t)}[c_1] - p_{v_t}^{(t)}[c_2]$

17: Inform the optimizer of $loss$

18: **end while**

19: $\mathbf{A}' \leftarrow \mathbf{A}, \mathbf{X}' \leftarrow \mathbf{X}$

20: Select $\eta_A^{(m)}, \eta_X^{(m)}$ that minimizes the $loss$

21: **for** each element $i$ in vector $\eta_A^{(m)}$ **do**

22: $\mathbf{A}'^{(t)}[v_t][i] \leftarrow 1 - \mathbf{A}[v_t][i]$

23: **end for**

24: **for** each element $j$ in vector $\eta_X^{(m)}$

25: $\mathbf{X}'^{(t)}[v_t][i] \leftarrow 1 - \mathbf{X}[v_t][i]$

26: **end for**

27: **return** $G' = (\mathbf{A}', \mathbf{X}')$

---

**Table 1 Dataset statistics.**

| Dataset | Nodes | Edges | Features | Classes |
|---|---|---|---|---|
| Cora | 2,708 | 5,429 | 1,433 | 7 |
| Citeseer | 3,327 | 4,732 | 3,703 | 6 |
| Polblogs | 1,490 | 19,025 | – | 2 |

**Table 2 Description of experimental parameters.**

| Parameters | Meaning | Range |
|---|---|---|
| Dataset | Dataset used to train the graph neural network | Cora, Citeseer and Polblogs |
| Target model[1] | The graph neural network model that will be attacked | 2-layer GCN, GCN-Jaccard and GCN-SVD |
| Node ID | Target nodes attacked in single-node experiments | $0 \sim N - 1$(only test set nodes are taken) |
| Number of target nodes | Number of target nodes for multi-node experiments | $0 \sim N_U - 1$ |
| Resource budget $\beta$ | The maximum number of iterations of the gradient-free optimizer. This parameter controls the number of computational resources | Positive integers |
| Perturbation type | Perturbation of structure or of features | Structure, feature, both |
| Constraint size $\Delta$ | The maximum number of edges or features that can be modified for each node manipulated by the attacker; this parameter controls the strength of the perturbation. | For structural perturbations: $1 \sim N$ |
| For feature perturbations: $1 \sim d$ | | |
| Scenario | Control whether the final perturbation is injected before training(poisoning) or after training (escape) | Poisoning or Evasion |
| DFOer | Indicates which derivative-free optimizer in Nevergrad is selected | OnePlusOne, DiscreteOnePlusOne, DoubleFastGA, etc. |

**Notes:**

[1] The same hyperparameters are chosen for the target model 2-layer GCN, GCN-Jaccard and GCN-SVD: hidden layer dimension is 16, dropout rate is 0.5, learning rate is 0.1, and weight decay is $5 \times 10^{-4}$.

## Setting

We describe some of the parameters and their range of values during the experiment in Table 2. The poisoning attack considered in this paper refers to the case where the model is not retrained at query time and is retrained at test time after the attack. In this case, our approach is equivalent to generating an adversarial sample with the training parameters of the original model and then transferring this sample to the model with the new parameters after retraining.

We define "original class" as the maximum probability class of the target node's black-box query before the attack. We define "successful attack" as the case that the maximum probability class after the attack is different from the original class. We define "misleading success" as the maximum probability class of the node after the attack is the same as the selected target class. "Classification margins" here are defined as $Z_{v, c} - \max_{c' \neq c} Z_{v, c'}$ where $c$ is the original class, $Z_{v, c}$ is the probability of the class $c$ given to the node $v$ by the attacked model. The lower the classification margins, the better the attack performance.

Besides DeepRobust, we also use packages include: Python 3.7; PyTorch 1.8.1; Nevergrad 0.4.3 post2; H5py 3.2.1, etc. In terms of hardware, the processor used for the experiment was a 2.6 GHz hexa-core Intel Core i7.

### Single-node Experiments (RQ1)

In this part, we attack just one node with different DFOers. We conducted Experiment 1 in order to find an appropriate DFOer by comparing their attack loss curves and

**Table 3 Parameter settings of Experiment 1.**

| Dataset | Model[1] | Node ID | $\beta$ | Type[2] | $\Delta$ | Scenario |
|---|---|---|---|---|---|---|
| Cora | 2-layer GCN | 0 | 100 | Structure | 5 | Poisoning |

**Notes:**
[1] Refers to "target model".
[2] Refers to "perturbation type".

**Table 4 Results of Experiment 1.**

| DFOer | Origin class | Target class | New class | Result | Margin |
|---|---|---|---|---|---|
| OnePlusOne | 5 | 6 | 6 | MS[1] | −0.67 |
| DiscreteOnePlusOne | 5 | 6 | 6 | MS | −0.75 |
| DoubleFastGA[2] | 5 | 6 | 6 | MS | −0.70 |
| DE | 5 | 6 | 6 | MS | −0.65 |
| RandomSearch | 5 | 6 | 6 | MS | −0.67 |

**Notes:**
[1] Means "misleading success".
[2] The full name is "DoubleFastGAOnePlusOne".

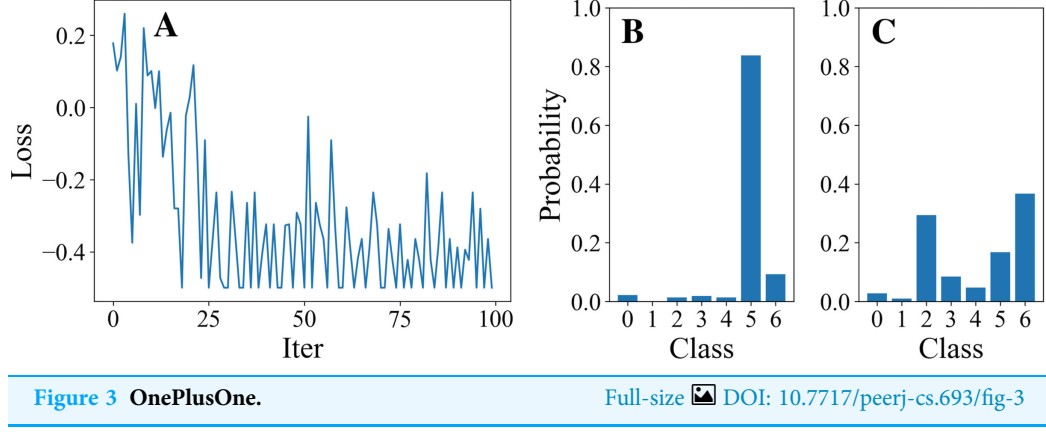

Figure 3 OnePlusOne.                

classification margins. In this experiment, different DFOers are taken to attack node 0 of Cora to let the model classify the node as class 6 instead of its original class 5.

## EXPERIMENT 1: ATTACK BY 5 DIFFERENT DFOERS ON NODE 0 OF CORA

Tables 3 and 4 show the parameter settings and results of Experiment 1 respectively. Figures 3 to 7 show the loss curves and class probability results of the 5 different DFOers. The figures with mark "A" show the attack loss curves of different DFOers. The horizontal coordinate is the number of iterations while the vertical coordinate is the value of the loss function. The smaller the loss function value, the greater the difference between the target class probability value and the original class probability value, the better the attack effect. The figures marked "B" and "C" show the class probabilities before and after the attacks. The horizontal coordinate is class labels, and the vertical coordinate refers to the probability that the node belongs to a certain class. The figures marked "B" shows the classification probabilities on unperturbed graphs and the figures marked "c" shows the

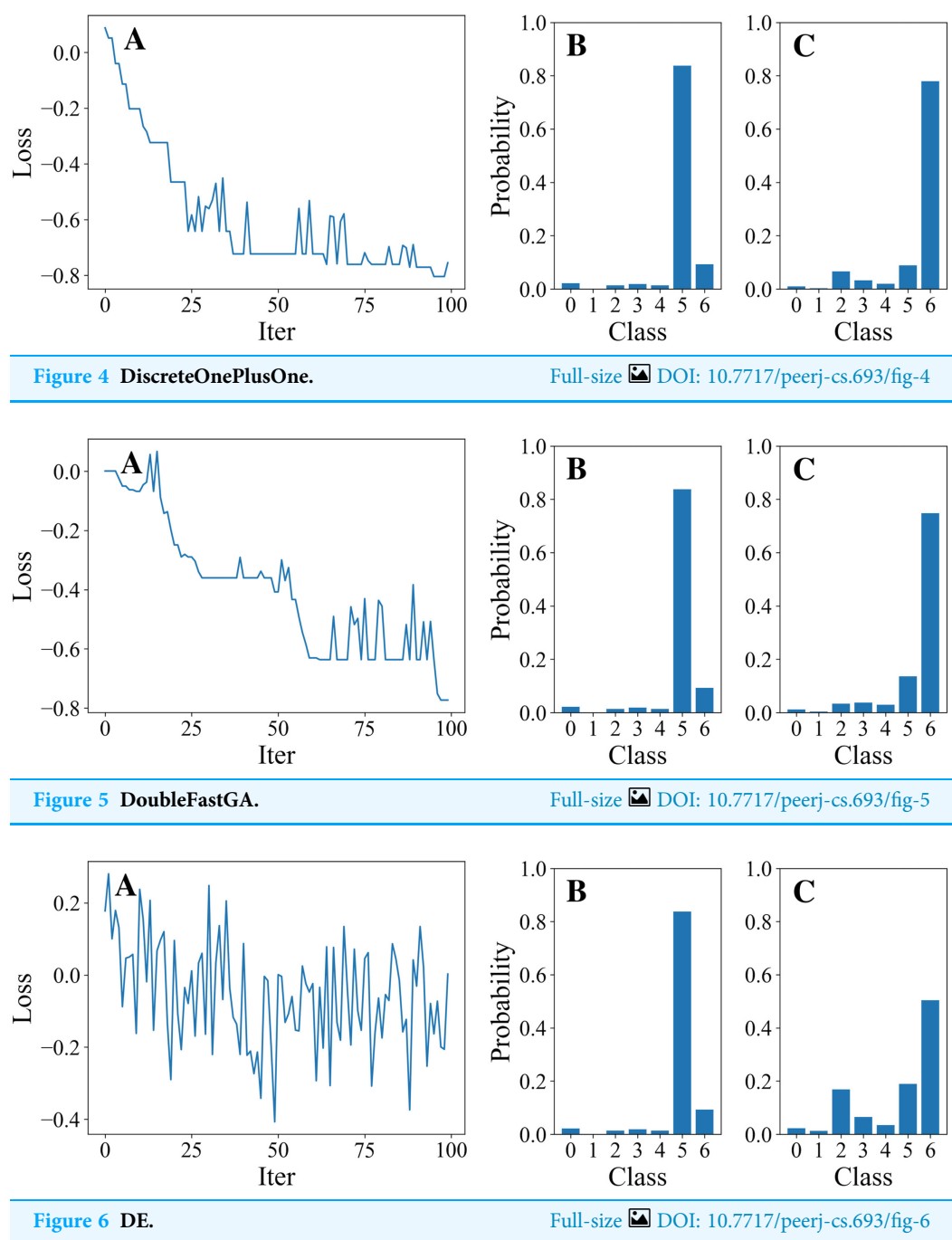

Figure 4 DiscreteOnePlusOne.               

Figure 5 DoubleFastGA.                     

Figure 6 DE.                               

probabilities on perturbed graphs. In this experiment, the greater the probability of class 6 and the smaller the probability of class 5, the better the experiment result is.

From the experimental result figures, we can see that various DFOers can all successfully mislead the model in our algorithm. As shown in Figs. 3A, 6A and 7A, OnePlusPne, DE, and RandomSearch can all obtain minimum values of −0.4 ~ −0.6, which can achieve less than −0.65 of the classification margin. However, judging from the oscillation degree of the attack loss curves, they do not have the potential of continuous optimization and

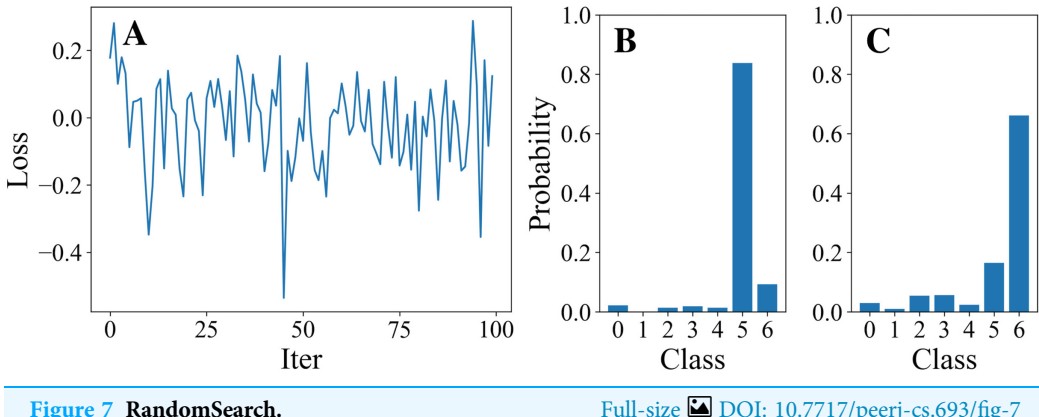

**Figure 7 RandomSearch.**

**Table 5 Parameter settings of Experiment 2.**

| Dataset | Number[1] | Type | Δ | Model | Scenario | DFOer |
|---|---|---|---|---|---|---|
| Cora | 50 | Structure | 5 | 2-layer GCN | Poisoning | DoubleFastGA |

Note:
[1] Number of the target nodes.

cannot continuously reduce the value of the loss function with the increase of iteration number.

From Figs. 4A and 5A, it can be seen that DiscreteOnePlusOne and DoubleFastGA can quickly find the appropriate optimization direction and optimize the loss function stably, and can reach a minimum value of about −0.8, corresponding to less than −0.7 of the attack power. However, during the running stage, DiscreteOnePlusOne often generates perturbations that violate the constraints, which is very inefficient. DoubleFastGA can combine speed and accuracy, so this optimizer is selected for the later experiments.

### Multi-node Experiments (RQ2)

In this part, we choose a set of target nodes from the test set and attack them separately. Every time we attack a node, we will test and record whether the attack was successful. We define the success rate (SR) as the ratio of the "successful attack" number to the total attacks number, Misleading Rate (MR) as the ratio of the "misleading success" number to the number of the total attacks. In Experiment 2 & 3, we try to find the effect of different parameter settings on SR and MR.

## EXPERIMENT 2: THE EFFECT OF DIFFERENT RESOURCE BUDGET β

The parameter settings and results of Experiment 2 are shown in Tables 6 and 5. The SR directly reflects the attack effectiveness of the algorithm: the higher the SR, the better the effectiveness. The MR reflects the directional misleading ability of the attack: the higher the ratio of MR to SR, the more directional the attack is.

The SR gradually increases with the increase of resource budget (*i.e.*, the number of iterations), but the time consumption is proportional to the number of iterations. The ratio

**Table 6  Results of Experiment 2.**

| β | SR (MR) | MR/SR | Running time | Running time (improved) |
|---|---------|-------|--------------|-------------------------|
| 10 | 0.68 (0.54) | 79.41% | 5 m 3 s | 3 m 39 s |
| 25 | 0.82 (0.72) | 87.80% | 9 m 15 s | 5 m 28 s |
| 50 | 0.80 (0.72) | 90.00% | 18 m 33 s | 10 m 30 s |
| 100 | 0.94 (0.86) | 91.49% | 37 m 11 s | 18 m 37 s |
| 200 | 0.94 (0.86) | 91.49% | 1 h 14 m 37 s | 33 m 35 s |

**Table 7  Parameter settings of Experiment 3.**

| Dataset | Number | Δ | Model | Scenario | DFOer |
|---------|--------|---|-------|----------|-------|
| Cora | 50 | 5 | 2-layer GCN | Poisoning | DoubleFastGA |

of MR to SR increases with the resource budget, which indicates that the DoubleFastGA is capable to find the appropriate optimization direction to make it more probable to classify nodes to the target class. When $\beta = 100$, the attack reaches the balance of efficiency and performance. It can be seen that even with the smallest resource budget ($\beta = 10$), 68% of the nodes are attacked successfully and 54% are misled successfully. When $\beta \geq 25$, the SR can exceed 80%.

During the experiment, we found that each iteration needs to perturb at the original image, which requires many deep copy operations on the original image which consumes a lot of computational time. We use "inverse perturbation" to solve this problem. During each iteration, we directly perturb the original graph data, and then "inversely perturb" the graph after each query to restore it to the original data. This approach greatly reduces the computational time of deep copies for a large number of iterations. As shown in the last two columns of Table 5, the time consumption after improvement is about 50% of that before.

## EXPERIMENT 3: INFLUENCE OF CONSTRAINT Δ AND PERTURBATION TYPE ON ATTACK EFFECT

Table 7 shows the parameter settings of Experiment 3. It can be seen from Table 8 that the structure perturbation effect is better than the feature perturbation. The combination of structure perturbation and feature perturbation can achieve a stronger attack effect but requires a more tolerant constraint (the sum of structural constraint and feature constraint). Only structure perturbations were used in subsequent experiments.

As the perturbation constraint size increases, the optimizer can search in a larger space. Obviously, a larger constraint size leads to a better attack effect.

### Comprehensive Experiments (RQ3)

In this part we conduct Experiment 4 & 5 to find how DFDA performs under 3 different models: base model 2-layer GCN mentioned in Eq. (2), GCN-Jaccard (*Wu et al., 2019*) and

**Table 8** Results of Experiment 3.

| Δ | Feature | Structure | Both[1] |
|---|---|---|---|
| 2 | 0.10 (0.08)[2] | 0.60 (0.56) | 0.70 (0.68) |
| 4 | 0.14 (0.14) | 0.88 (0.84) | 0.92 (0.88) |
| 6 | 0.22 (0.20) | 0.96 (0.86) | 0.94 (0.94) |
| 8 | 0.24 (0.24) | 0.92 (0.80) | 0.98 (0.80) |
| 10 | 0.30 (0.30) | 0.98 (0.92) | 0.98 (0.94) |

**Notes:**
[1] When structure perturbations and feature perturbations are carried out simultaneously, each type of perturbations has a constraint of Δ.
[2] Represents "SR(MR)".

**Table 9** Parameter settings of Experiment 4.

| Dataset | Number | Type | β | Scenario | DFOer |
|---|---|---|---|---|---|
| Cora | 50 | Structure | 100 | Evasion | DoubleFastGA |

**Table 10** Results of Experiment 4.

| Δ | 2-layer GCN | GCN-Jaccard | GCN-SVD |
|---|---|---|---|
| 2 | 0.46 (0.46) | 0.63 (0.53) | 0.63 (0.60) |
| 4 | 0.86 (0.80) | 0.83 (0.76) | 0.80 (0.73) |
| 6 | 0.86 (0.83) | 0.93 (0.83) | 0.90 (0.76) |

GCN-SVD (*Entezari et al., 2020*). Experiment 4 is conducted in an evasion scenario where the model doesn't retrain, while Experiment 5 is in a poisoning scenario that needs retraining before attacks. Moreover, we compare DFDA with classic method *Nettack* in Experiment 5. We randomly attack 30 test set nodes with DFDA (using the DoubleFastGA optimizer) and Nettack respectively. We repeated the process five times and calculated the mean and standard deviation of the SRs to make the comparison more plausible. Both Experiment 4 & 5 are conducted on Cora, Citeseer and Polblogs.

# EXPERIMENT 4: EFFECTIVENESS OF DFDA UNDER DIFFERENT DEFENSIVE MODELS IN EVASION ATTACK SCENARIO

Parameter settings of Experiment 4 are shown in Table 9. As we can see from Table 10, DFDA can achieve good results in the evasion scenario. In this scenario, both of the defense models cannot effectively defend against adversarial samples. This may be due to the fact that the defense mechanisms (*e.g.*, graph purification mechanism of GCN-Jaccard and low-rank approximation processing of GCN-SVD) are essentially pre-processing of the input data. As the model does not retrain, the defense mechanisms are bypassed in our evasion setup. This why the success rate on the defense models is abnormally high.

**Table 11 Parameter settings of Experiment 5.**

| Number | Type | β | Scenario | DFOer |
|---|---|---|---|---|
| 30 | Structure | 100 | Poisoning | DoubleFastGA |

**Table 12 Results of Experiment 5.**

| Dataset | β | 2-layer GCN | | GCN-Jaccard | | GCN-SVD | |
|---|---|---|---|---|---|---|---|
| | | DFDA | Nettack | DFDA | Nettack | DFDA | Nettack |
| Cora | 2 | 62.7 ± 9.2 | **69.3 ± 3.7** | 39.3 ± 7.2 | **42.7 ± 14.0** | **20.0 ± 7.8** | 10.0 ± 6.7 |
| | 4 | **85.3 ± 6.9** | 78.7 ± 10.2 | 58.0 ± 9.3 | **61.3 ± 11.5** | **48.6 ± 11.9** | 22.7 ± 6.4 |
| | 6 | **95.3 ± 6.9** | 88.0 ± 3.8 | **73.3 ± 9.4** | 72.0 ± 12.2 | **68.0 ± 5.6** | 42.7 ± 9.5 |
| | 8 | **95.3 ± 3.8** | 85.3 ± 3.8 | **80.0 ± 8.5** | 79.3 ± 6.4 | **67.3 ± 5.5** | 56.0 ± 15.3 |
| Citeseer | 2 | **70.0 ± 14.5** | 53.3 ± 9.7 | **54.7 ± 9.6** | 39.3 ± 8.6 | **41.3 ± 7.7** | 32.0 ± 8.7 |
| | 4 | **83.3 ± 7.5** | 73.3 ± 6.2 | **72.0 ± 7.7** | 61.3 ± 5.1 | **65.3 ± 10.2** | 54.7 ± 9.0 |
| | 6 | **88.7 ± 8.0** | 68.0 ± 6.1 | **81.3 ± 6.5** | 70.0 ± 4.1 | **73.3 ± 5.3** | 68.7 ± 5.1 |
| | 8 | **98.0 ± 1.8** | 74.0 ± 6.4 | **88.0 ± 8.0** | 77.3 ± 9.8 | **78.7 ± 6.5** | 72.0 ± 6.9 |
| Polblogs | 2 | **51.3 ± 5.1** | 28.0 ± 9.6 | **41.3 ± 21.0** | 31.3 ± 10.2 | **7.3 ± 4.3** | 1.3 ± 1.8 |
| | 4 | **56.7 ± 10.3** | 54.0 ± 2.8 | **44.7 ± 17.7** | 30.0 ± 4.7 | **13.3 ± 6.7** | 6.7 ± 5.3 |
| | 6 | **69.3 ± 2.8** | 58.0 ± 3.8 | **41.3 ± 21.9** | 32.0 ± 3.8 | **23.3 ± 10.3** | 14.0 ± 6.0 |
| | 8 | **70.0 ± 7.8** | 64.7 ± 9.6 | **39.3 ± 21.1** | 35.3 ± 8.7 | **34.6 ± 6.9** | 31.3 ± 17.2 |

Note:
*The data represent the means and standard deviations of SRs. The bold indicates the average SR of the algorithm that performs better under the same conditions.

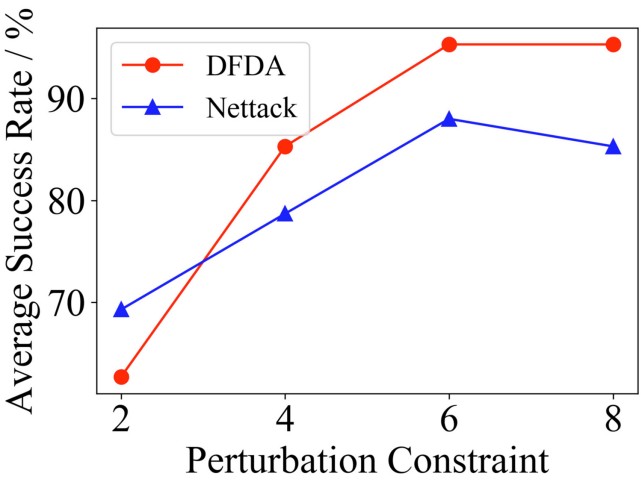

**Figure 8 Average SR on 2-layer GCN & Cora.**

# EXPERIMENT 5: COMPARISON OF DFDA AND *NETTACK* IN DIFFERENT DEFENSE MODELS UNDER POISONING ATTACK SCENARIOS

Table 11 shows the parameter settings of Experiment 5. The results are shown in Table 12 and are visualized in nine line graphs from Fig. 8 to 16. With $\beta = 100$ and $\Delta = 8$, DFDA

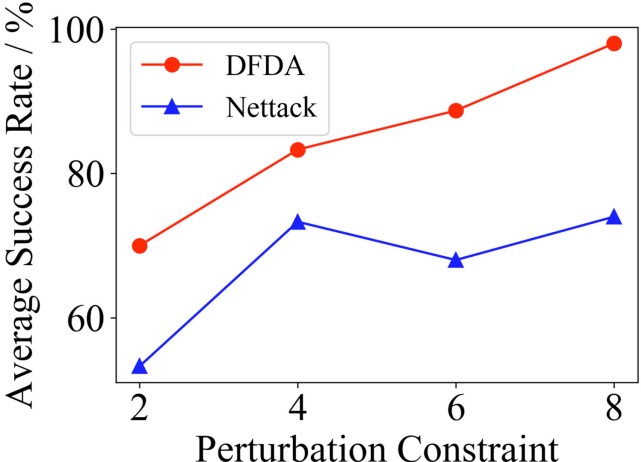

**Figure 9  Average SR on 2-layer GCN & Citeseer.**   

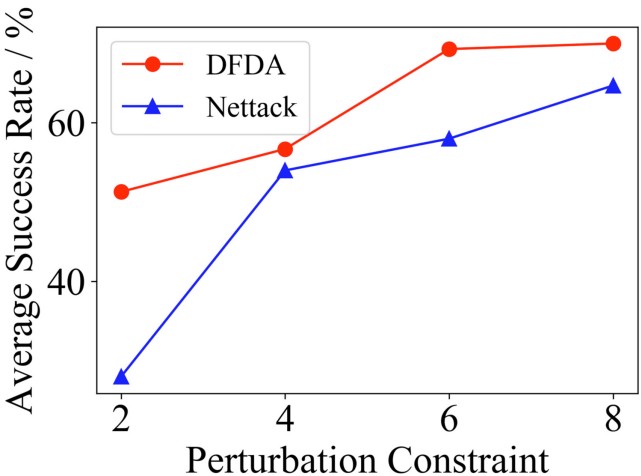

**Figure 10  Average SR on 2-layer GCN & Polblogs.**   

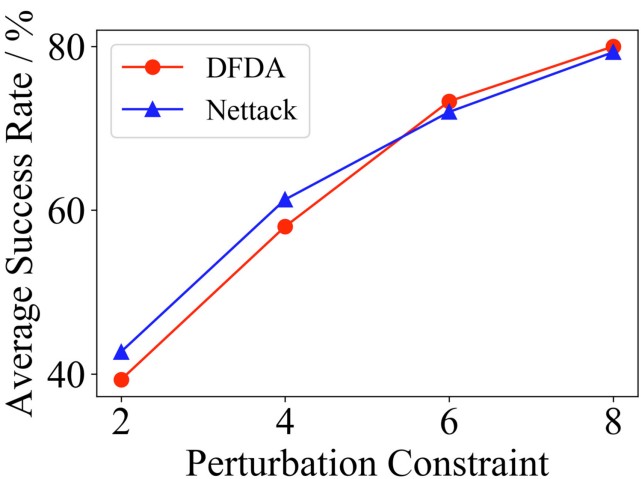

**Figure 11  Average SR on GCN-Jaccard & Cora.**   

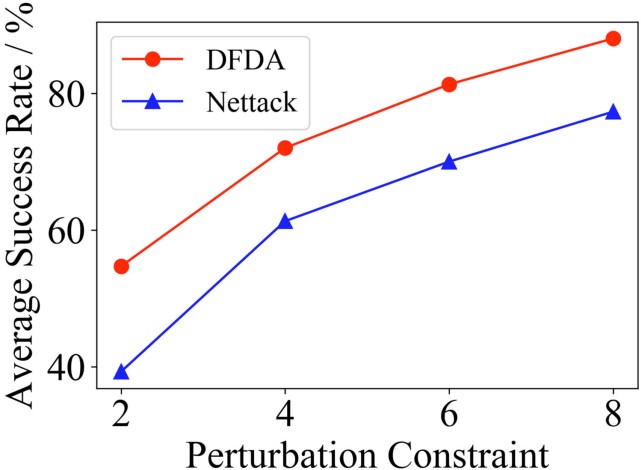

**Figure 12 Average SR on GCN-Jaccard & Citeseer.**

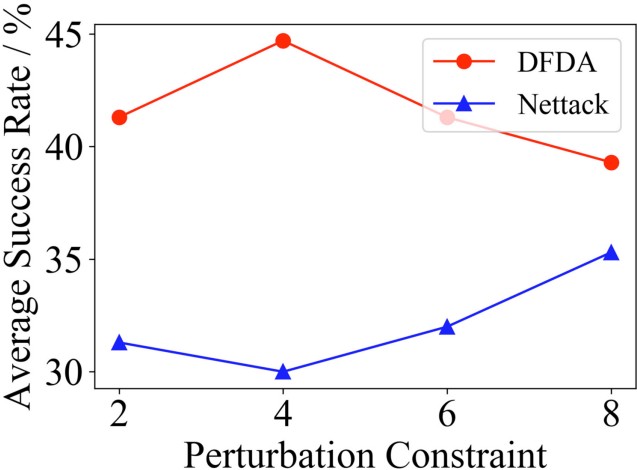

**Figure 13 Average SR on GCN-Jaccard & Polblogs.**

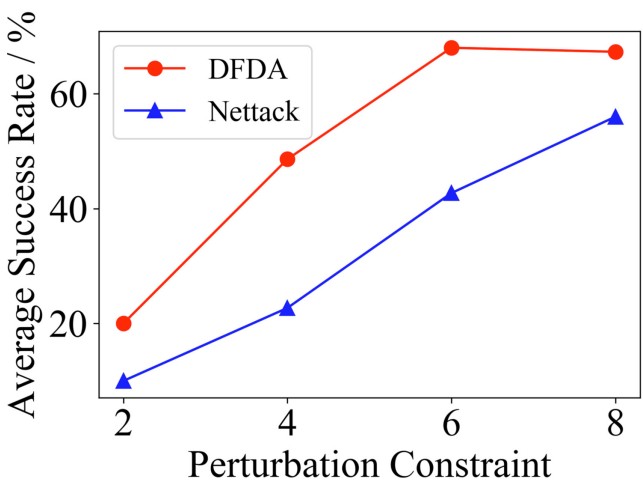

**Figure 14 Average SR on GCN-SVD & Cora.**

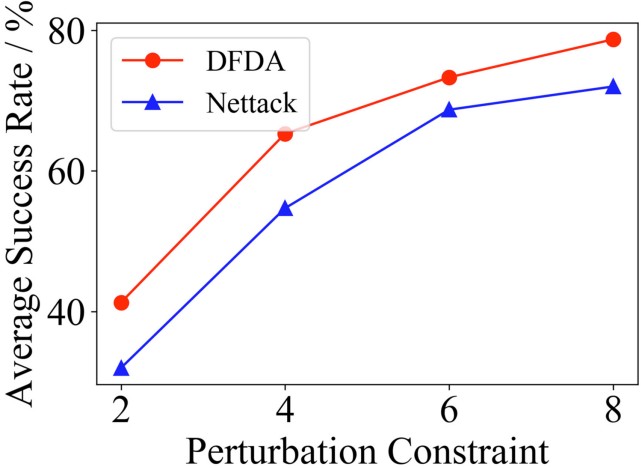

**Figure 15** **Average SR on GCN-SVD & Citeseer.**

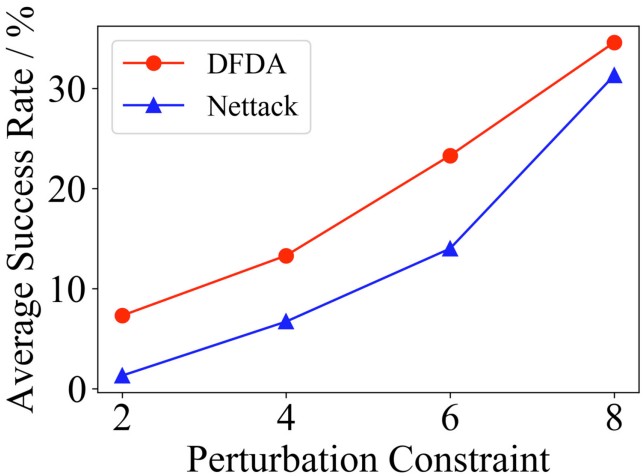

**Figure 16** **Average SR on GCN-SVD & Polblogs.**

achieves a maximum average SR of 95.3%, 98.0% and 70.0% on Cora, Citeseer and Polblogs, respectively. DFDA performs more excellent on Cora and Citeseer than on Polblogs. This is because there are only two class labels in Polblogs and the edges are denser than Cora and Citeseer, which would make Polblogs more difficult to attack.

The performance of DFDA on three models: 2-layer GCN > GCN-Jaccard > GCN-SVD. The GCN-Jaccard is less defensive against DFDA. This may be because the adversarial perturbations generated by the DFDA algorithm do not easily reduce the Jaccard similarity between nodes. So it is not easy to be defended by GCN-Jaccard. GCN-SVD is more defensive against DFDA because the variation of high-rank singular components in the adversarial sample spectrum generated by DFDA is large. Therefore, the perturbations generated by DFDA will be easily filtered by the matrix approximation algorithm.

DFDA generally outperforms Nettack on three datasets and three target models. Nettack uses training labels to train the surrogate model and generates adversarial samples by attacking it. This algorithm uses greedy ideas in optimization and attacks much faster,

but it cannot optimize iteratively. DFDA can increase the number of iterations to raise the SR by spending more time. The time complexity of Nettack is related to the constraint size ($O(\Delta)$), while the time complexity of DFDA is related to the number of iterations ($O(\beta)$).

Of the parameters involved in the above experiments, the three parameters—scenario, perturbation type and target model—are used to distinguish between different types of adversarial attacks, in decreasing order of importance. The scenario distinguishes between poisoning attacks and evasion attacks, the perturbation type distinguishes between structure and feature perturbations, and the target model distinguishes between attacks under different defense models.

There are three numerical type parameters that influence the effectiveness of the attack. In descending order of importance, they are constraint size, resource budget and number of target nodes. Constraint size determines the number of the edges or features that can be manipulated and directly controls the ease of the attack task. The resource budget determines the number of iterations. A large resource budget can increase the success rate of the attack to some extent. The number of target nodes determines the stability of the success rate of the attack. The larger the nodes number, the less the fluctuation in success rate.

## CONCLUSION

In this paper, we focus on the use of derivative-free optimization (DFO) ideas in graph adversarial attacks. We first introduce a DFO-based black-box adversarial attack framework against GCNs. Then we implement a direct attack algorithm (DFDA) using Nevergrad library, using which we can easily compare the performance of various derivative-free optimizers on node classification attack tasks. Moreover, we solve the problem of large search space by declining the perturbation vector dimension. Finally, we conducted three kinds of experiments on Cora, Citeseer and Polblogs. The results show that DFDA outperforms Nettack in most cases. It can achieve an average success rate of more than 95% on Cora when perturbing at most eight edges, which demonstrates that our method can fully exploit the potential of DFO methods in node classification adversarial attacks. In the future, we will focus on the application of DFO ideas to indirect attacks and global attacks.

### Funding
This work was supported by the National Natural Science Foundation of China (No. 62002332). The funders had no role in study design, data collection and analysis, decision to publish, or preparation of the manuscript.

### Grant Disclosures
The following grant information was disclosed by the authors:
National Natural Science Foundation of China: 62002332.

## Competing Interests

The authors declare that they have no competing interests.

## Author Contributions

- Runze Yang conceived and designed the experiments, performed the experiments, analyzed the data, performed the computation work, prepared figures and/or tables, authored or reviewed drafts of the paper, and approved the final draft.
- Teng Long conceived and designed the experiments, analyzed the data, authored or reviewed drafts of the paper, and approved the final draft.

## Data Availability

All the source code files, datasets and a readme file are available in the Supplementary Files.

## Supplemental Information

Supplemental information for this article can be found online at http://dx.doi.org/10.7717/peerj-cs.693#supplemental-information.

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
