# Peer review of "Derivative-free optimization adversarial attacks for graph convolutional networks"

_PeerJ Computer Science, doi:10.7717/peerj-cs.693_

## Round 0.1 · original submission · Minor Revisions

The main topic of the paper is of special interest. The structure and content are OK. A few minor improvements are proposed by reviewers.

Reviewer 1 ·

Basic reporting

This research article proposed a black-box adversarial attack framework based on Derivative-Free Optimization (DFO) and a direct attack algorithm (DFDA) using Nevergrad library to classify adversarial attacks on graph nodes. Authors have used Cora, Citeseer and Polblogs data sets for the experimental evaluation. The proposed work achieved an average attack success rate of more than 95%.

Experimental design

The introduction part is well written. The contribution of the proposed work has been highlighted precisely. Authors are suggested to shift the related work after the Introduction part. Related work is well written. Proposed work is well defined using the algorithm. The dataset used for the experimental evaluation is also well explained. All the results obtained are well explained by using various evaluation parameters. Authors are suggested to include future work in the conclusion section.

Authors have used the experimental parameters, namely, target model, node ID, number of target nodes, resource budget, perturbation type, constraint size, scenario, and DFOer. Authors are suggested to give the order of these parameters based on their importance towards classifying adversarial attacks on graph nodes, after the experimental evaluation.

Validity of the findings

This research article proposed a black-box adversarial attack framework based on Derivative-Free Optimization (DFO) and a direct attack algorithm (DFDA) using Nevergrad library to classify adversarial attacks on graph nodes. The paper is well written. The proposed methodology, data set, and results discussed are well written. The paper is suitable to be published in this journal after incorporating minor changes suggested.

Additional comments

No Comments

Reviewer 2 ·

Basic reporting

The authors studied adversarial attacks to Graph Convolutional Networks. As a result, the authors proposed a black-box adversarial attack framework based on derivative-free optimization and implemented a direct attack algorithm. The results showed that their proposal outperformed the baseline method Nettack.

The manuscript is literature supported.

This paper is nicely written, well-organized, and easy to read.

- Line 43 > uppercase
- Line 235 > “DeeoRobust” - DeepRobust
- Figure 3, 4, 5 and 6 should be improved. In overall the labels are too small.

Experimental design

Source code, examples and data sources were provided.

Python was used as programming language, which helps to reach a bigger audience.

The authors performed an extensive experimental study comparing both performance and execution time, which is welcome.

Validity of the findings

The authors provided an extended discussion about the results and their proposal won the 91% of the time by an acceptable margin in Table 12.

The proposal reduced the execution time in almost the half.

---

## Round 0.2 · accepted · Accept

This new version addressed satisfactorily the comments provided by the reviewers, so the paper is acceptable for publication.

Reviewer 1 ·

Basic reporting

The authors have addressed the comments suggested.

Experimental design

Authors were suggested to include future work. Also authors were suggested to give the order of the parameters based on their importance towards classifying adversarial attacks on graph nodes, after the experimental evaluation.

Authors have included the future work and mentioned about the importance of parameters.

Validity of the findings

The paper may be accepted in the current state.

Additional comments

I congratulate and appreciate the efforts of the authors in preparing the manuscript. I wish them very best for their future research work.

Reviewer 2 ·

Basic reporting

The authors have improved the manuscript following the suggestions made by the reviewers.

The manuscript is acceptable in its current form.

Experimental design

-

Validity of the findings

-